# Novel Selective and Low-Toxic Inhibitor of *Lm*CPB2.8ΔCTE (CPB) One Important Cysteine Protease for *Leishmania* Virulence

**DOI:** 10.3390/biom12121903

**Published:** 2022-12-19

**Authors:** Vitor Partite Moreira, Michele Ferreira da Silva Mela, Luana Ribeiro dos Anjos, Leonardo Figueiredo Saraiva, Angela M. Arenas Velásquez, Predrag Kalaba, Anna Fabisiková, Leandro da Costa Clementino, Mohammed Aufy, Christian Studenik, Natalie Gajic, Alexander Prado-Roller, Alvicler Magalhães, Martin Zehl, Ingrid Delbone Figueiredo, Amanda Martins Baviera, Eduardo Maffud Cilli, Marcia A. S. Graminha, Gert Lubec, Eduardo R. Perez Gonzalez

**Affiliations:** 1Fine Organic Chemistry Lab, School of Sciences and Technology, São Paulo State University (UNESP), Presidente Prudente 19060-080, Brazil; 2School of Pharmaceutical Sciences, São Paulo State University (UNESP), Araraquara 14800-903, Brazil; 3Laboratory of Luminescence in Materials and Sensors, School of Sciences and Technology, São Paulo State University (UNESP), Presidente Prudente 19060-560, Brazil; 4Department of Pharmaceutical Sciences, Division of Pharmaceutical Chemistry, Faculty of Life Sciences, University of Vienna, Josef Holaubek Platz 2, UZAII, 1090 Vienna, Austria; 5Mass Spectrometry Centre, Faculty of Chemistry, University of Vienna, Währinger Straße 38, 1090 Vienna, Austria; 6Department of Pharmaceutical Sciences, Division of Pharmacology and Toxicology, University of Vienna, Josef Holaubek Platz 2, UZAII (2D 259), 1090 Vienna, Austria; 7Centre for X-ray Structure Analysis, Faculty of Chemistry, University of Vienna, Währinger Straße 40-42, 1090 Vienna, Austria; 8Department of Organic Chemistry, Chemistry School, Federal University of Rio de Janeiro, Rio de Janeiro 21941-598, Brazil; 9Department of Analytical Chemistry, Faculty of Chemistry, University of Vienna, Währinger Straße 38, 1090 Vienna, Austria; 10Department of Biochemistry and Organic Chemistry, Institute of Chemistry, São Paulo State University (UNESP), Araraquara 14800-060, Brazil; 11Department of Neuroproteomics, Paracelsus Medical University, 5020 Salzburg, Austria

**Keywords:** *Leishmania* cysteine protease inhibition, guanidines, X-ray and NMR conformational study, molecular docking, leishmanicidal activity

## Abstract

Leishmaniasis is a highly prevalent, yet neglected disease caused by protozoan parasites of the genus *Leishmania*. In the search for newer, safer, and more effective antileishmanial compounds, we herein present a study of the mode of action in addition to a detailed structural and biological characterization of **LQOF-G6** [*N*-benzoyl-*N*′-benzyl-*N*″-(4-tertbutylphenyl)guanidine]. X-ray crystallography and extensive NMR experiments revealed that **LQOF-G6** nearly exclusively adopts the *Z* conformation stabilized by an intramolecular hydrogen bond. The investigated guanidine showed selective inhibitory activity on *Leishmania major* cysteine protease *Lm*CPB2.8ΔCTE (CPB) with ~73% inhibition and an IC_50-CPB_ of 6.0 µM. This compound did not show any activity against the mammalian homologues cathepsin L and B. **LQOF-G6** has been found to be nontoxic toward both organs and several cell lines, and no signs of hepatotoxicity or nephrotoxicity were observed from the analysis of biochemical clinical plasma markers in the treated mice. Docking simulations and experimental NMR measurements showed a clear contribution of the conformational parameters to the strength of the binding in the active site of the enzyme, and thus fit the differences in the inhibition values of **LQOF-G6** compared to the other guanidines. Furthermore, the resulting data render **LQOF-G6** suitable for further development as an antileishmanial drug.

## 1. Introduction

Leishmaniasis is a neglected disease present in 98 countries and is caused by more than 20 species of the parasitic protozoan genus *Leishmania* (Kinetoplastide: Trypanosomatidae), which has a higher prevalence and incidence in Southeast Asia, sub-Saharan Africa, and Latin America [1]. The disease includes a wide spectrum of clinical-pathological manifestations such as cutaneous (CL), mucocutaneous (MCL), or visceral leishmaniasis (VL), the latter form being in 85–90% of untreated cases [2].

It is estimated that up to 12 million people are currently infected worldwide while hundreds of millions live in high-risk areas [3,4]. Leishmaniasis is common in Brazil, and since the 1990s, there has been a significant increase in the number of cases [1]. The difficulty in controlling the phlebotomine vector as well as domestic reservoirs along with the lack of accurate diagnosis has increased the number of deaths [4].

Cutaneous leishmaniasis, caused by *Leishmania amazonensis*, is the most common form of the disease, causing skin lesions and ulcers. Mucocutaneous leishmaniasis devastates the mucous membranes of the nose, mouth, and throat [5,6,7,8]. Visceral leishmaniasis, being the most severe form of the disease, is characterized by anemia, significant loss of weight, increased spleen, fever, and often death [9,10,11,12,13].

Among the treatment methods for the disease, the most popular, leishmaniasis chemotherapy, is based on either the use of pentavalent antimonial drugs that were developed decades ago (meglumine antimoniate, trade name Glucantime, and sodium stibogluconate, sold under the brand name Pentostam) [14], or amphotericin B (Amp B), paromomycin, pentamidine, and miltefosine, all of which present severe side effects, require long periods of treatment, have low efficacy, and have inadequate routes of administration [15,16]. Pentavalent antimonials, despite still being the first line of intervention, are rather old drugs administered intramuscularly with highly toxic effects [17]. Amp B, pentamidine salts, paromomycin sulfate, and oral miltefosine represent the second line of drugs. They also exhibit toxic effects with the necessity of repeated doses over a long period [17]. Thus, the search for newer, safer, and more effective antileishmanial compounds is essential.

On that account, selectively targeting the enzymes that are necessary for the functional integrity and virulence of *Leishmania* may lead to drugs with a defined mode of action and no off-side effects. Studies have shown the key role of the parasite’s cysteine proteases in the infection process and suggested the high therapeutic potential of inhibiting these enzymes, enabling the design of new drugs [18]. Considering *L. mexicana*, three genes of cysteine protease were identified as virulence factors, namely *lmcpa*, *lmcpb,* and *lmcpc* [19]. Of these, *lmcpb* codes for a protein product that is cathepsin L-like and is the most abundant in the amastigote form of the parasite [19].

Regarding the previous information on the inhibition of cysteine proteases of the specific type *Lm*CPB, one study reported a benzo[*b*]thiophene derivative ((4aS,9bS,9cS)-4-methyl-9c-(4-methylbenzoyl)-4a-phenyl-4,4a,9b,9c-tetrahydro-1H-benzo[4,5]thieno[2,3-b]furo[3,4-d]pyrrole-1,3(3aH)-dione) as a potent and selective inhibitor of *Lm*CPB2.8△CTE (CPB), with an IC_50-CPB_ of 3.7 µM [2], (Figure 1a). Also reported was a guanidine, which targets cysteinyl proteases, the compound E64 (L-t*rans*-epoxysuccinyl-leucylamido-(4-guanidino)butane)**,** an irreversible and potent inhibitor of *Leishmania* CPB at nanomolar range [20], (Figure 1b). Unfortunately, E64, like other inhibitors of cysteinyl proteases, shows toxic effects or interference with human cathepsins L and B. Furthermore, to be considered for therapeutic purposes and drug development, inhibitors need to exert in vivo activity.

Different factors may affect the inhibition of *Lm*CPB2.8ΔCTE, including the interaction mechanism between the ligand and the acceptor (enzyme). The key to the clinical effect of the drugs lies in the binding of their bioactive components and the corresponding targets to the exertion of their pharmacological activities [21]. Additionally, knowledge of the system is essential, not only for predicting the binding but also for understanding the pharmacodynamics and pharmacokinetics of the interaction [22]. To elucidate and comprehend the main mechanisms, computational techniques are often employed, considering that they have been exponentially enhanced in the last decade, to accomplish simulations in desktop computers [23]. Among the several methods, molecular docking is considered the state-of-the-art computational procedure in drug construction and has become a major computer-aided drug design (CADD) method, since it can effectively predict the binding mode and energy of the protein-ligand complexes [21].

In the previous study, a series of guanidines, **LQOF-G1**, **-G2**, and **-G6** has been reported and their pharmacological activity has already been addressed [24]. The new compound **LQOF-G32** was also selected for enzymatic *Leishmania Lm*CPB2.8ΔCTE inhibition experiments, based on its good in vitro activity and high safety index (SI) values, namely: **G1** (36.45), **G2** (123), **G6** (65.82), and **G32** (27.27). In addition, **LQOF-G32** presents structural differences in the benzylic ring as shown in Figure 2.

The most potent enzymatic inhibitor compound (**LQOF-G6**) was investigated for cell and organ toxicity, and leishmanicidal activity in vivo, using female BALB/c mice infested with *L*. *amazonensis*. Furthermore, two compounds were selected for a conformational study - the one that performed the best in inhibiting the *Lm*CPB2.8ΔCTE enzyme (**LQOF-G6**) and another that did not show any inhibition (**LQOF-G1)** - to differentiate them and understand the reason for this difference in inhibition using biological, NMR, XRD, and computational methods. Thus, to get a better understanding of how compounds with inhibitory activity interact with the CPB enzyme, a quantum mechanical calculation-docking study was performed for all the investigated compounds and the results were in accordance with the experimentally determined inhibition data.

## 2. Materials and Methods

### 2.1. Synthesis of Guanidines

(*Z*)-*N*-benzoyl-*N*′-benzyl-*N*″-(4-tertbutylphenyl) guanidine **LQOF-G6**, **LQOF-G1** –(4-nitrophenyl), and **LQOF-G2**-(4-bromophenyl) guanidines were synthesized as previously reported [24]. The **LQOF-G32** (*Z*)-*N*-benzoyl-*N*′-(2,4-dichlorobenzyl)-*N*″-(4-bromophenyl)guanidine is a new guanidine that was prepared using the same procedure.

General synthesis was performed by the reaction between thioureas (specific for each guanidine) with aniline. Thiourea is solubilized in DMF, triethylamine is added and Bi(NO_3_)_3_·5H_2_O is added sequentially. The reaction time is 24 h at a temperature of 120 °C under magnetic agitation and reflux. After the end of the reaction, the reaction mixture is filtered through celite and extracted with dichloromethane, washed with water, and the organic phase is separated, dried on magnesium sulfate, filtered, and the residual product deposited after the evaporation of the solvent, is recrystallized.

### 2.2. Structural Characterization

#### 2.2.1. NMR Measurements

The ^1^H, ^13^C, and various 2D NMR in solution spectra were acquired in a Bruker AVANCE-III HD—500 MHz instrument at 253 K if not otherwise stated. Chemical shifts (δ) are referenced using tetramethylsilane (TMS). The analyses were performed with 15 mg of guanidines solubilized in 500 µL of CDCl_3_ 99.8 atom % D, containing 0.03% (*v*/*v*) TMS. The ^1^H data are presented as follows: chemical shifts, multiplicity (s = singlet, d = dublet, dd = double dublet, t = triplet, dt = double triplet, tt = triple triplet, qua = quartet, qu = quintet, m = multiplet, br s = broad singlet), coupling constants (J) in Hertz (Hz) and peak integrals. Solid state NMR experiments were carried out in a triple channel Bruker AVANCE NEO—500.13 MHz for ^1^H, using a 4 mm CRAMPS zirconia rotor.

#### 2.2.2. Single Crystal X-ray Diffraction Analysis

Crystal data, data collection parameters, and structure refinement are summarized in the Appendix A). The X-ray intensity data were measured on a Bruker D8 Venture equipped with a multilayer monochromator, Mo K/α microfocus sealed tube, and an Oxford cooling system. The structure was solved by the *Dual Space algorithm*. Nonhydrogen atoms were refined with *anisotropic displacement parameters*. Hydrogen atoms were inserted at calculated positions and refined with a riding model. Additionally, to the software from Bruker listed in the cif file following programs were used: *OLEX2* for structure solution, refinement, molecular diagrams, and graphical user interface [25]; *Shelxle* for refinement and graphical user interface [26]; *SHELXS-2015* for structure solution [27]; *SHELXL-2015* for refinement [28]; Platon for symmetry check [29].

### 2.3. Biological Toxicity Studies

#### 2.3.1. Cell Toxicity

The human embryonic kidney cell line HEK-293 (obtained from ATCC) as well as three human cancer cell cultures, the epidermal carcinoma-derived cell KB-3-1 (generously donated by Dr. Shen, Bethesda, MD, USA), the colon carcinoma cell line CaCo-2 and the adenocarcinomic human alveolar basal epithelial cell A549 (obtained from ATCC) were used in this study. Cells were cultivated in Dulbecco’s modified Eagle medium (DMEM, Gibco by Life Technologies, LifeTech Austria, Vienna, Austria), supplemented with 5% fetal bovine serum (FBS, Gibco by Life Technologies, LifeTech Austria) and 1% penicillin-streptomycin (Sigma-Aldrich, Vienna, Austria) at 37 °C with 5% CO_2_ in a humidified incubator. Cultures were periodically checked for *Mycoplasma* contamination. Cells were seeded in 96 well plates at a density of 2 × 10^4^ cells/mL in 100 µL per well and allowed to attach for 24 h. Afterward, cells were incubated with 100 µL of **LQOF-G6** diluted in DMEM at concentrations ranging from 1.56 to 400 µM. Stock solutions were prepared in water with 10% (*v*/*v*) dimethyl sulfoxide (DMSO) and stored at 4 °C. The proportion of viable cells was determined after 72 h exposure to **LQOF-G6** by 3-(4,5-dimethylthiazol-2-yl)-2,5-diphenyltetrazolium bromide (MTT)-based vitality assay kit (EZ4U, Biomedica, Vienna, Austria) [30,31]. Briefly, 20 µL of the EZ4U assay solution was added to each well. After 2 h of incubation in the dark, the absorbance was measured by a microplate reader, at 450 nm with 620 nm as a reference to reduce unspecific background values. All experiments were performed three times in triplicates.

#### 2.3.2. Organ Toxicity on Isolated Tissue Preparations

Guinea pigs of either sex weighing 250–380 g have been used. Animals were kept in an air-conditioned room at a temperature of 22–24 °C and relative humidity of 50–60% with a 12-h photo period. On the day of the experiments, the animal was sacrificed by cervical dislocation. The heart, aorta, pulmonary artery, and ileum were surgically excised and kept in a Krebs-Henseleit solution (NaCl 144.9 mM, KCl 4.73 mM, CaCl_2_ 3.2 mM, MgSO_4_ 1.18 mM, NaHCO_3_ 24.9 mM, KH_2_PO_4_ 1.18 mM and glucose 10 mM; pH 7.2–7.4), continually aerated with 95% O_2_ and 5% CO_2_. Papillary muscles were dissected from the right ventricle of the heart and cleared from Purkinje fibers to avoid spontaneous activity. We used muscles having a diameter of less than 0.87 mm to ensure proper oxygen supply. The right atrium was also dissected to check the chronotropic activity. Both the aorta and the pulmonary artery were cleaned, and rings of 5 mm were cut while the ileum was cut from the terminal portion into pieces of 1–2 cm. One end of the dissected tissues was tied with silver wire for attachment to the tissue holder while the other end was attached to the force transducer (Transbridge, 4-Channel Transducer Amplifier, World Precision Instruments, Sarasota, FL, USA). The terminal ileum was contracted with 60 mM KCl while a 90 mM KCl solution was used for the pulmonary artery and aorta rings. Papillary muscles were electrically stimulated by an Anapulse Stimulator model 301-T and an Isolation UnitModel 305-1 (WPI, Hamden, CT, USA) with rectangular pulses of 3 ms at a frequency of 1 Hz. The amplitude of the stimulation pulse was kept 10% above the threshold level. To obtain maximum contractility from the respective tissues, a constant resting tension of 3.9 mN for papillary muscle, 4.9 mN for terminal ileum, 10.4 mN for the right atrium, and 19.6 mN for aorta and pulmonary artery rings was applied throughout the experiment. After a control period of 30 min, different concentrations (3, 10, 30, and 100 µM) of test compounds were applied cumulatively in a bath solution every 45 min until a steady effect was obtained. The responses were recorded by a chart recorder (BD 112 Dual Channel, Kipp & Zonen) and evaluated later. Stock concentrations for the test compounds were made with DMSO. To exclude the effect of DMSO, experiments were performed with solvent only, and the observed effect was subtracted from the response of the test compounds. Data are presented as means ± SEM and were statistically evaluated using Student’s *t*-test for paired observations. A *p*-value of less than 0.05 was considered to indicate a significant change.

### 2.4. Inhibitory Activity on the Enzyme LmCPB2.8ΔCTE (CPB)

The recombinant **CPB**, lacking the *C*-terminal extension, was expressed and the inhibitory assay was carried out according to a reported procedure [20]. Briefly, in a black 96 well-plate containing sodium acetate buffer 100 mM (pH 5.5), 200 mM NaCl, 0.01% Triton-X100, 2 nM of the enzyme, and 2 mM dithiothreitol (DDT), with a volume of 196 μL, the enzyme was pre-incubated at 37 °C for 2 min before addition of 2 μL of the guanidine **LQOF-G1**, **LQOF-G2**, **LQOF-G6**, and **LQOF-G32** in different concentrations (1.25–20 μM) and finally 2 μL of the substrate, 5 μM Z-Phe-Arg-7-amido-4-methylcoumarin (Z-FR-AMC, Sigma-Aldrich) were added. The amount of 7-amino-4-methylcoumarin released by **CPB** was monitored by recording its fluorescence (380 nm excitation and 460 nm emission filters) for 2 min in the Infinite 200 PRO microplate reader (Tecan, Männedorf, Switzerland) followed by calculation of the residual activity, given as the ratio of the background-subtracted fluorescence after incubation with **LQOF-G6** to the one without the addition of the guanidine. The results were expressed in mean ± standard deviation of two independent replicates using Bioestat Software as the half maximal inhibitory concentration of the enzyme **CPB** (IC_50-CPB_).

#### 2.4.1. Enzyme Kinetic Assay

The kinetic experiments for **CPB** were carried out as previously described with some modifications as follows [20]. To a solution containing 100 mM acetate buffer (pH 5.5), 200 mM NaCl, 0.01% *v*/*v* Triton X-100, and 2 mM DTT was added 7 nM **CPB** at 37 °C using Corning 96-well black flat bottom microplates. The enzyme stock aliquot was rapidly thawed at 37 °C and kept on ice until activation, in which it was incubated for 5 min in the assay buffer followed by an additional 5 min incubation with an inhibitor (**LQOF-G6**) before the reaction was started by the addition of several concentrations of the substrate Z-FR-AMC to give a final volume of 200 µL. The rate of the reaction was monitored for a total of 5 min using an Infinite 200 PRO microplate reader (Tecan) through the fluorescence emission at 460 nm (excitation at 380 nm) which is proportional to the hydrolysis of the substrate. The initial rates of substrate hydrolysis under first-order reaction conditions were calculated by GraphPad Prism software based on the linear-regression coefficient from the data fitting of the relative fluorescence unit (mRFU) as a function of time (min). Each experiment was performed in duplicates. A control measurement in absence of an inhibitor was carried out for each setup plate. The inhibitor was evaluated at three different concentrations (3 μM, 6 μM, and 10 μM), which were chosen based on the IC_50-CPB_.

#### 2.4.2. Proteases Assays

A study of **LQOF-G6** on the inhibition of human Cathepsin L and Cathepsin B was outsourced to Eurofins (Cerep SA, Celle l’Evescault, France, Study No. 100062592). For more information, please refer to the Appendix A.

### 2.5. Leishmanicidal Activity

#### 2.5.1. Parasites

*Leishmania amazonensis* promastigotes (MPRO/BR/1972/M1841-LV-79) were maintained at 28 °C in liver-infusion tryptase medium (LIT) supplemented with heat-inactivated fetal bovine serum (iFBS), streptomycin (10 mg/mL), and penicillin (1000 U/mL) [24,32,33].

#### 2.5.2. Antileishmanial In Vivo Assay

In vivo leishmanicidal activity of guanidine **LQOF-G6** was performed as previously described [24,34]. Briefly, female BALB/c mice (20 to 25 g; 5 weeks old; CEMIB, UNICAMP) were subcutaneously inoculated at the right ear with ~10^7^ infective promastigotes at the stationary growth phase of *L. amazonensis*. After 60 days postinfection and lesion growth to 5 to 7 mm in diameter, the animals were randomly separated into four groups containing five animals each (Table 1). The group of treated animals received either 2 mg/kg (body weight)/day of **LQOF-G6** or Amp B (reference drug) for 15 days through intraperitoneal administration (IP). **LQOF-G6**-animal doses were calculated based on IC _50-ama_ previously determined [18] and the animal’s blood volume (1.5 mL) which refers to 6% of the animal’s weight. The final **LQOF-G6**-animal dose is 10-fold of IC_50-ama_. Stock solutions of **LQOF-G6** were prepared daily in 1X PBS after solubilization in DMSO (final concentration, 0.1%). A group of infected animals treated with 1X PBS (vehicle) via IP and another group containing uninfected and untreated mice (healthy animals) served as controls. The course of infection during the treatment was monitored three times a week by measuring the thickness and the diameter of the lesions using a dial caliper (Mitutoyo Corp., Japan). The efficacy of the treatment was determined by measuring the parasite burden of the infected ears by LDU index analyses [33].

#### 2.5.3. Toxicity Assay for BALB/c Mice

Plasma levels of total bilirubin, aspartate aminotransferase (AST), alanine aminotransferase (ALT), alkaline phosphatase (ALP), urea, and creatinine were determined in blood plasma samples of BALB/c mice at the end of the treatments using commercial kits (Labtest Diagnostica S.A., Brazil) as previously described [24,33].

#### 2.5.4. Statistical Analysis

The statistical differences between groups were evaluated using the one-way analysis of variance (ANOVA) test, followed by Tukey’s multiple comparisons test (using GraphPad InStat software). Differences were considered significant when *p* ≤ 0.01.

#### 2.5.5. Ethics Statement

Animal experiments were approved by the Ethics Committee for Animal Experimentation of São Paulo State University (UNESP), the School of Pharmaceutical Sciences (CEUA/FCF/CAr n° 32/2020) in agreement with the guidelines of the Brazilian Society of Laboratory Animal Science (SBCAL) and the National Council for the Control of Animal Experimentation (CONCEA).

### 2.6. Docking Investigation

First, for the docking procedure, the ligand was prepared using the LigPrep software (implemented in the Maestro molecular modelling environment). The structures of **LQOF-G1** and **LQOF-G6** were obtained from single-crystal XRD data, while **LQOF-G2** and **LQOF-G32** molecules were from a quantum mechanical geometry optimization at the density functional theory (DFT) level using the Def2-TZVP basis set [34] and B3LYP functional [35] through the Orca 5.0.3 software [36]. The target was prepared using BIOVIA Discovery Studio with the following procedure: (i) obtaining the. pdb (protein data bank file) target structure (6P4E); (ii) removal of water molecules, subunit B, cosolvent, metals, and cocrystalized ligands; (iii) removal of the H atoms and posterior addition of polar H from the software; (iv) selection of the most favorable positions of the amino acids; (v) optimization of the system. The docking grid was generated using BIOVIA Discovery Studio, applying a sphere in the center of the CYS26 residue with sufficient size to accommodate the 20 Å ligands, with (−0.4442, 12.943, 46.679) for (x, y, z) coordinates. The docking was performed with AutoDock Vina 1.2.0 software [37] using PDBQT files through the AutoDock Tools 1.5.7 interface [38]. During the calculations, the Lamarckian genetic algorithm was chosen, in which the ligand was kept flexible whereas the protein was kept rigid to obtain different conformations with several binding energies [39]. The docking output file was saved in the same format containing all the interaction information, where nine interacting conformers were spawned with different ligand-target affinity. Among the results, the file with the highest affinity was chosen to be visualized and analyzed using BIOVIA Discovery Studio.

## 3. Results and Discussion

### 3.1. Structural Characterization

In the process of this work, the structural characterizations of the molecules **LQOF-G6** and **LQOF-G32** were performed, initially by in-solution NMR, whose results are described in the Appendix A. The molecules **LQOF-G1** and **LQOF-G2** were previously reported [24].

Based on the biological data (which are presented below in Section 3.2), two molecules were subjected to a conformational investigation by coupling NMR data with single-crystal XRD. The selected molecules were **LQOF-G1** and **LQOF-G6**, which showed opposing results of inhibitory activity towards the enzyme *Lm*CPB2.8ΔCTE, with **LQOF-G1** demonstrating no inhibition, while **LQOF-G6** showed the highest activity as an inhibitor of CPB.

These molecules, in addition to having opposite activities against the enzyme, were also selected due to their complete crystallographic studies. This study was carried out with the aim of finding the preferential conformation in the solid state. Table 2 shows the experimental crystallographic parameters for **LQOF-G6** and **LQOF-G1**.

Figure 3 shows the most stable conformations for both compounds, **LQOF-G1** and **LQOF-G6**. A detailed description of the conformation is described in the SI.

Before describing and discussing the differences in the activity of these molecules, the biological data observed for **LQOF-G6,** with better inhibiting activity, are presented below. We assume that the torsion of the aniline ring plays an important role in the inhibiting activity as further discussed in the docking investigation (Section 3.4).

### 3.2. Biological Assays

#### 3.2.1. Cell and Organ Toxicity

Figure 4A shows the normalized cell viability of four different cell lines upon treatment with **LQOF-G6** at a concentration range of 1.56–400 µM. No significant effect on the cell viability was observed, even at the highest concentration values, demonstrating the absence of cytotoxicity of **LQOF-G6** for all four cell lines, in accordance with our previous work that shows the low toxicity of **LQOF-G6** to macrophages, the primary resident cell for *Leishmania* (half-maximal cytotoxicity concentration > 1 mM) [24].

It is noteworthy to emphasize the link between the nontoxic effect of **LQOF-G6** toward the human embryonic kidney cell line HEK-293 and the lack of kidney toxicity, as suggested by the unaltered creatinine levels in mouse blood plasma (see below).

The inotropic and chronotropic effects of the compound **LQOF-G6** were evaluated in the isolated heart muscle preparations of guinea pigs in a concentration range between 3 and 100 µM. **LQOF-G6** did not cause a significant change in the aortic rate of activity up to a concentration of 30 µM. At a concentration of 100 µM, **LQOF-G6** slightly decreased the rate of activity (Figure 4B). The contractibility in the papillary muscle was not affected by concentrations up to 100 µM, (Figure 4B).

The vasodilating effect was studied in the aortic and arteria pulmonalis rings, and the spasmolytic effect in the terminal ilea. The preparations were stimulated with 90 mM KCl (aortic and arteria pulmonalis rings) and 60 mM KCl (terminal ilea). There was no significant effect on the aorta (Figure 4C) up to a concentration of 100 µM.

However, **LQOF-G6** had a clear effect on the arteria pulmonalis and at the final tested concentration of 100 µM, the contraction force was decreased by 41.75 ± 7.93% in arteria pulmonalis rings (Figure 4C).

In the terminal ilea, **LQOF-G6** also concentration-dependently relaxed the KCl-induced contraction force (f_c_). At the highest concentration studied (100 µM), the compound reduced the f_c_ from 8.71 ± 1.35 to 2.93 ± 1.00 mN (*n* = 5, *p* < 0.01). The IC_50_ value for the spasmolytic effect was determined to be 14 µM (Figure 4C).

**LQOF-G6** does not have cardiac side effects in isolated heart muscle preparations up to concentrations of 100 µM. The vasodilating activity on the aortic rings is not significant, yet there is a decrease in the contraction force in the arteria pulmonalis rings. Guanidine derivatives are selective iNOS inhibitors and high doses can partially inhibit the constitutive eNOS isoform [40,41]. This activity might be the reason for the observed vasodilation of the pulmonary artery. In the terminal ilea, a significant spasmolytic effect was detected with an IC_50_ value of 14 µM. This might be due to the antihistaminergic effects reported for guanidine derivatives [42].

In addition to the graphs presented here, the Appendix A contains the tables detailing the effect of **LQOF-G6** on each organ analyzed (Appendix A).

#### 3.2.2. Inhibition of *Leishmania* Cysteine Protease

*Leishmania* resides in the phagolysosome mononuclear phagocytic cells, particularly in macrophages. To evade the host’s immune response, the parasite uses several mechanisms, including CPB secretion, to modulate the host’s Th1/Th2 response, favoring its proliferation [43]. Studies have shown that some guanidine derivatives can inhibit the activity of this enzyme and are therefore interesting prototypes for the design of new antileishmanial drugs targeting the CPs [44,45].

This work describes the experimental kinetic study of the inhibition of *Lm*CPB2.8ΔCTE protease by **LQOF-G6**. The percentage of inhibition of the enzyme by **LQOF-G6** was initially investigated at a concentration of 20 μM. **LQOF-G6** showed a 73% inhibitory activity against CPB with IC_50-CPB_ of 6.0 ± 0.2 μM. This is a highly relevant result, which differentiates **LQOF-G6** from other guanidines of this series, such as the previously reported **LQOF-G2 [24]**. **LQOF-G2** is a potent leishmanicidal compound in vivo against *L. amazonensis*; however, it did not show inhibitory activity against *Lm*CPB2.8ΔCTE.

Furthermore, using 20 μM of the substrate, **LQOF-G6** was found to be a competitive inhibitor of *Lm*CPB2.8ΔCTE at its IC_50-CPB_ value (6.0 ± 0.2 μM) with a V_max_ = 500 μM/min and K_m_ = 25.3 μM. In fact, **LQOF-G6** showed increased values of K_m_ with increasing concentrations from 3 ± 0.2 μM, to 6 ± 0.2 μM and 10 ± 0.2 μM, which aligns with a competitive behavior (Figure 5).

Although other natural and synthetic CPB inhibitors have been reported [20], none of them have been preclinically followed up on in cutaneous leishmaniasis models. It is worth mentioning the selectivity of **LQOF-G6**, which did not inhibit human cathepsins B and L at concentrations ranging from 2 to 10 μM (Appendix A).

These results suggest that **LQOF-G6** can be proposed for more advanced studies as a candidate for therapeutic agents against *L. amazonensis* infections.

The enzymatic inhibition of human isoforms of cathepsin L and cathepsin B was essayed by Eurofins. **LQOF-G6** was tested at 1 µM and 10 µM concentrations, and no significant inhibition (>50%) was observed for either of the tested concentrations. The whole report can be found in the Appendix A.

The novel **LQOF-G32** molecule was also tested as a CPB inhibitor and showed 53% of inhibition at the test concentration of 20 µM, which is somewhat lower compared to **LQOF-G6**. With the promising data of enzymatic inhibition of **LQOF-G6** in hand, leishmanicidal activity was still tested to complement the work.

#### 3.2.3. Leishmanicidal Activity

##### Reduction of Parasite Load by **LQOF-G6** in BALB/c Mice Infected with *L. amazonensis*

Based on previous data showing the in vitro leishmanicidal activity of **LQOF-G6** against intracellular amastigote forms of *L. amazonensis* (EC_50ama_ = 17 µM) and its safety (SI = 65) [24], the in vivo potential of **LQOF-G6** to diminish the parasite load in BALB/c mice infected with *L. amazonensis* was evaluated here.

The infected mice were treated daily with 2.0 mg/kg/day of **LQOF-G6** or Amp B for 15 days (Figure 6). As a result, the animals treated with **LQOF-G6** presented significantly decreased lesion sizes (~46%) at the end of the treatment compared with the group treated with the vehicle (Figure 6A,C). Moreover, **LQOF-G6** decreased the number of parasites at the site of the lesion by 80% compared to the vehicle, and it was even more effective than Amp B (Figure 6B). Additionally, it can be highlighted that the leishmanicidal activity was also tested for the **LQOF-G32** molecule, with values of 14 ± 1.2 and 11 ± 1.4 µM for IC_50-pro_ and IC_50-ama_, respectively.

##### Investigation of Hepatic or Renal Disturbance

Changes in hepatic and renal function and damage were monitored in *L. amazonensis*-infected BALB/c mice after the treatment with **LQOF-G6** or Amp B, including the vehicle for comparison, by measurement of the plasma levels of creatinine, urea (biomarkers of renal function), total bilirubin (a biomarker of hepatic function), ALP, ALT, and AST (biomarkers of hepatic damage) (Figure 7). Overall, the few observed alterations in the **LQOF-G6** group are likely due to the disease rather than to the treatment. Changes in creatinine and urea levels are associated with the impairment of renal function [46]. Although the levels of creatinine did not change in either of the evaluated groups (Figure 7A), the urea levels for both **LQOF-G6** and Amp B were higher than those for the healthy animals (Figure 7B). This might be due to the disease rather than the treatment since the levels of this marker were also high in the infected animals treated with PBS. Indeed, changes in renal markers caused by CL have been previously reported [24,46].

**LQOF-G6** also did not change total bilirubin (Figure 7C), ALP (Figure 7D), ALT, and AST (Figure 7E,F) plasma levels compared with infected animals.

Amp B is an antifungal agent that interacts with ergosterol, which is also a component of the parasite cell membrane, causing the pore formation that makes the cell membrane permeable for cations, anions, and some metabolites, ultimately leading to cell death [47]. However, the emergence of resistant strains affects Amp B efficacy in leishmaniasis chemotherapy. Additionally, its severe toxic effects involve renal failure, fever, bone pain, and cardiac arrest, among others, as well as contributing to low patient compliance [48,49]. Thus, the search for newer, safer, and more effective antileishmanial molecules is essential.

Our data suggest that **LQOF-G6** is very efficient in the CL treatment caused by *L. amazonensis* due to its higher capacity to decrease the parasite load compared to Amp B. Additionally, **LQOF-G6** showed no toxicological effects, making it a good candidate for CL treatment and a promising candidate for further exploitation of treatment leishmaniasis clinical manifestations.

After evidencing the biological data, emphasizing the inhibition capacity towards the enzyme *Lm*CPB2.8ΔCTE that **LQOF-G6** showed, the conformational study of this molecule was initiated in comparison to **LQOF-G1**, which did not present enzyme inhibition, while still showing in vitro activity against *L. amazonensis.* Thus, both compounds, **LQOF-G6** and **LQOF-G1**, were studied by XRD.

### 3.3. Conformational Analysis

The X-ray study of **LQOF-G1** and **LQOF-G6** has shown their crystal structure with the double bond of the guanidinium group located between the central carbon (C1A in DRX and C7 in NMR) and the amide N (N2A in DRX and Nc in NMR), and the intramolecular hydrogen bond [50] between the hydrogen atom on the aniline N (N1A in DRX and Nb in NMR) and the carbonyl oxygen atom (O1A in DRX) was identified by its direct connection to Na (90.25 ppm) and showed one long-distance correlation with the Nc nitrogen (172.51 ppm). H4 (7.21 ppm) was correlated with Nb (105.54 ppm). H2 (12.09 ppm) also correlated with Nb (105.54 ppm) by a direct connection, and with Na (90.25 ppm); H1 (direct link) and H13, with chemical shifts of 5.33 and 4.83 ppm, respectively, correlated with Na (90.25 ppm). NMR in solution data showed that compound **LQOF-G6** is almost quantitively in the ***Z*** conformation. The occurrence of the ***E*** isomer was not demonstrated by NMR analyses (for more information please refer to the Appendix A).

As observed in the ^1^H spectrum at 253 K shown in **SI**, the signals of protons H2 and H1 appeared as singlet and triplet, respectively, as can be expected. However, this information is gradually lost at higher temperatures due to peak broadening. Recording the NMR spectra at 253 K to obtain sharper peaks and thus the higher resolution allowed determination of the multiplicity of H1 and H2 and thus further confirming the assignment of these signals. With H2 now unequivocally identified, the strong downfield shift of the signal is evidence for its involvement in the hydrogen bond with the carbonyl group and only confirmed the *Z* isomer as the predominant one.

Generally, the **LQOF-G1** molecule showed the same interactions as **LQOF-G6**, due to the similarity of the structures. In the presence of a different group in the para position of the aniline ring, the chemical shifts are not the same, having some changes.

#### 3.3.1. Investigation of Spatial Coupling by NOESY

Another interesting feature of the **LQOF-G6** molecule, which was observed by X-ray diffraction, was the distortion of the plane containing the aniline ring in relation to the plane of the guanidine bond (Figure 3 in Section 3.1 above). The repulsive steric interaction that can occur between the hydrogen atoms bound to C7A and N3A, as well as to N1A and C3A, could be a stabilizing factor for the conformation with the distortion of the tertbutyl aniline ring. These torsion angle values are consistent with spatial distances between H(C3A)-H(N3A) and H(C7A)-H(N1A) being shorter.

Alternatively, **LQOF-G1** did not show the same distortion of the plane of the aniline ring relative to the plane of guanidinic bonds. The distance between these planes is small for **LQOF-G1**, thus showing a significant change in the dipole-dipole coupling signals of NOESY in relation to **LQOF-G6**. The values of torsion angles obtained from the XRD of the **LQOF-G1** and **LQOF-G6** molecules are shown in Table 3.

NOESY experiments were carried out to analyze the interactions of nuclei spatially close to each other. The NOESY spectra of **LQOF-G6** at 253 K (Appendix A) show the highlighted peaks H2-H4 and H1-H4, as well as the crossing peaks H13-H10 which refer to the correlation of nonequivalent nuclei important to the ***Z*** isomer of **LQOF-G6**.

Regarding the previously highlighted torsion angles, it was possible to observe that the NOESY signals of **LQOF-G6** present different intensities from **the LQOF-G1** signals, precisely due to the distortion of the aniline ring. For example, the distance from H(C7A)-H(N3A) for **LQOF-G6** is 3.43A, while for **LQOF-G1**, it is 1.88A. This difference propagates in NOESY analyses, as for **LQOF-G1** these hydrogens are closer, and the signal of this dipole-dipole correlation has a higher intensity than for **G6**.

In summary, the NMR spectra acquired at 253 K show highly informative signals to confirm the strong predominance of the ***Z*** isomer, which is stabilized by the hydrogen bond formation between the aniline NH and the carbonyl oxygen (NH-----O=C) resulting in a six-members pseudo cycle. The distortion of the *p*-tertbutyl aniline benzenic ring was shown by the X-ray and NOE correlation between the hydrogen atoms H(C7A) and H(N1A).

#### 3.3.2. Solid-state NMR

In addition to the NMR experiments in solution, the analysis of ^13^C signals of **LQOF-G6** in the solid state was performed. In agreement with the X-ray crystal structure, this analysis showed the predominance of the ***Z*** isomer also in the absence of CDCl_3_ as the solvent (Appendix A).

Compared to the ^13^C spectrum in solution (158.659 ppm, Appendix A), the guanidinic central carbon (C7) was observed with a chemical shift of 163.498 ppm. Another important feature in the structure of the guanidine molecule is the C13 (benzyl), which is observed at 48.660 ppm in the solid-state spectrum and 45.765 ppm in solution. Differences between the solution and solid-state data are also observed for carbons C9 and C14, which are observed at 140.893 and 141.863 ppm, respectively, in solid-state and at 138.175 and 138.562 ppm in solution. These relatively small differences in the chemical shift values are probably a consequence of the solvent effect rather than conformational changes.

### 3.4. Docking Investigation

The experimental results have shown that **LQOF-G6** possesses the highest inhibitory activity among the small series of guanidines, which can further be investigated by docking simulations. This section will focus on the **LQOF-G6** guanidine, and a complete report on the other compounds can be found in the Appendix A. The binding affinity of **LQOF-G6** with the main chain of the enzyme was −8.8 kcal.mol^−1^, which is a high value compared to those guanidines that do not show inhibitory activity, such as **LQOF-G1** (−7.9 kcal.mol^−1^). The interactions mainly responsible for these values are highlighted in Figure 8, which shows a comparison of the docking results of **LQOF-G1** and **LQOF-G6** in two- and three-dimensional perspectives.

As we can see, in the **LQOF-G1** guanidine the MET147, ALA143, and CYS26 residues possess the same type of interaction (π-alkyl) through π-orbitals. Alternatively, in **LQOF-G6,** a new interaction of the same type occurs with the aromatic ring (LEU162), whereas the affinity with the CYS26 residue increases.

From our investigation, the docking data indicate that the CYS26 residue plays a key role in boosting the binding affinity considering that the sphere was centered in this residue. From our quantum mechanical calculations, the electron-withdrawing −NO_2_ group changes the position of the π-molecular orbitals (MOs) in the ring, whereas the localization was found only in the LUMO-type orbitals, reducing the availability of these MOs to interact with the CYS26 residue because of the higher energies displayed by these π-orbitals. Thus, in the **LQOF-G6** compound, an opposite behavior was observed due to the methyl groups, shifting to lower energies of the MOs upon the ring. Nevertheless, the increase in this interaction can be assigned to the change of the −NO_2_ to the −CH_3_ group due to the electron-donating character of the methyl. The other compounds **LQOF-G2** and **LQOF-G32** were also investigated through the docking procedure as well, and Table 4 summarizes the obtained results for the affinity of the ligand-target binding related to the “best” mode of all compounds. The effectiveness of the **LQOF-G2** molecule has already been investigated from an experimental perspective [24], although we highlight that the electron-donating character through the resonance effect of bromine leads to downshifting in the energy position of the orbitals. This shift induces higher π-alkyl interactions with the active site than **LQOF-G1**, but not as high as the **LQOF-G6** molecule. The same behavior is observed in the new **LQOF-G32** molecule with the inclusion of chlorine atoms in the benzylic ring. Nevertheless, this ring does not actively participate in the binding; therefore, it does not effectively contribute to the increase of binding affinity.

From Table 3, the **LQOF-G2** molecule does not exhibit inhibition activity, although its ligand-protein affinity is higher than **LQOF-G32**. This anomalous behavior is assigned to the direct relation of this property with the Gibbs energy of binding ($\Delta G_{bind}$), which we describe in the limit of the Ajay and Murcko partition (Equation (1)) [51]. In this model $\Delta G_{solvent}$ is the solvation/desolvation individual energy, $\Delta G_{conf}$ is the change in the receptor and ligand energy due to the complex formation, $\Delta G_{int}$ is the change in energy caused by the specific interactions between the ligand and the receptor, and $\Delta G_{motion}$ is the contribution due to changes in the motion (rotation, translation, and vibration).
(1)$$\Delta G_{bind}=\Delta G_{solvent}+\Delta G_{conf}+\Delta G_{int}+\Delta G_{motion}$$

Assuming that $\Delta G_{int}$ is higher for **LQOF-G32** compared with **LQOF-G2** due to the inhibition index, the $\Delta G_{conf}+\Delta G_{motion}$ should be greater for the **LQOF-G2** molecule, considering the same $\Delta G_{solvent}$, leading to a higher Gibbs energy of binding and, consequently, higher binding affinity.

## 4. Conclusions

X-ray crystallography, extensive 1D and 2D NMR experiments in CDCl_3_, as well as ^13^C solid-state NMR data, proved that **LQOF-G6** nearly exclusively adopts the *Z* conformation stabilized by an intramolecular hydrogen bond. This is also observed by coupled DRX and NMR investigation for **LQOF-G1**.

**LQOF-G6** causes inhibitory activity on the *Lm*CPB2.8ΔCTE, with 73% at 20 μM with IC_50_ = 6.0 μM. This compound is a reversible, competitive, and selective inhibitor, without any activity against human cathepsin isoforms B and L. The computational data predict the interactions of this molecule with the CPB enzyme. The docking shows that this interaction is energetically more favorable for **LQOF-G6** than for **LQOF-G32**, **LQOF-G1**, and **LQOF-G2**. This explains the higher biological activity of **LQOF-G6**.

**LQOF-G6** showed no signs of cytotoxicity up to 200 µM, had no inotropic or chronotropic effect in isolated heart muscle preparations of guinea pigs up to 100 µM, and did uncritically influence the contraction force of the aorta. However, a weak vasodilating activity on arteria pulmonalis rings prepared from guinea pigs was observed and a significant spasmolytic effect was measured in the terminal ilea (IC_50_ = 14 µM), likely due to antihistaminergic effects.

Fitting to these findings, no evidence of a hepatic or renal disturbance was observed when monitoring several biomarker levels in the plasma of BALB/c mice infected with *L. amazonensis* and treated daily with 2.0 mg/kg/day **LQOF-G6** (IP) over 15 days. Most relevant, this treatment decreased the *L. amazonensis*-induced lesions to nearly half their size, and the number of parasites at the site of the lesion was reduced by 80% compared to the vehicle, being superior to Amphotericin B in the latter parameter.

The current treatment options for leishmaniasis, mainly chemotherapy, are far from ideal due to the frequent occurrence of significant side effects and the development of resistance to certain drugs. The presented data here suggest that **LQOF-G6** is a new, safe, and effective antileishmanial lead compound that should be further developed in more extensive preclinical studies.

## Figures and Tables

**Figure 1 biomolecules-12-01903-f001:**
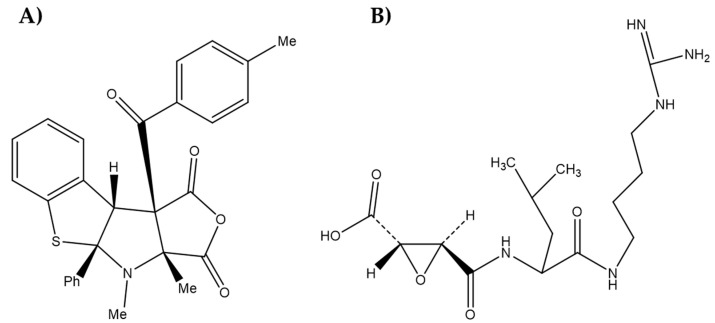
Structures of *Leishmania* CPB inhibitors: (**A**) (4aS,9bS,9cS)-4-methyl-9c-(4-methylbenzoyl)-4a-phenyl-4,4a,9b,9c-tetrahydro-1H-benzo[4,5]thieno[2,3-b]furo[3,4-d]pyrrole-1,3(3aH)-dione and (**B**) L-t*rans*-epoxysuccinyl-leucylamido-(4-guanidino)butane (E64).

**Figure 2 biomolecules-12-01903-f002:**
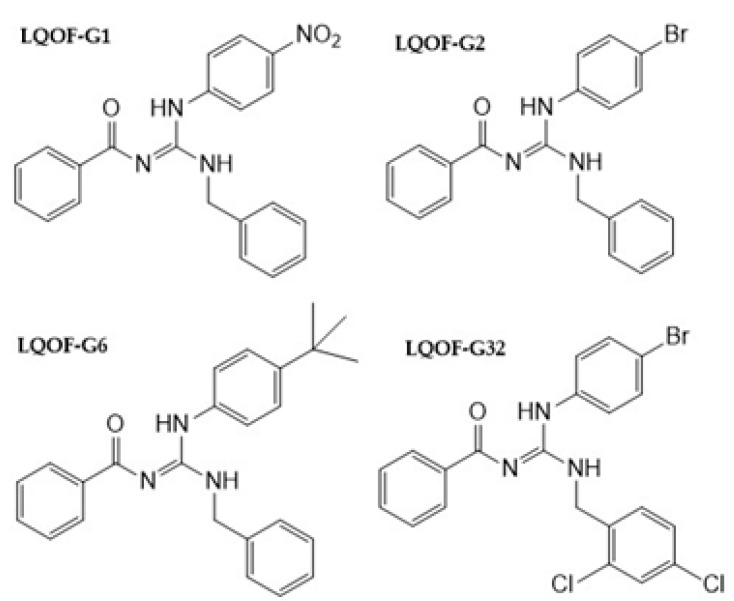
Structures of investigated guanidines.

**Figure 3 biomolecules-12-01903-f003:**
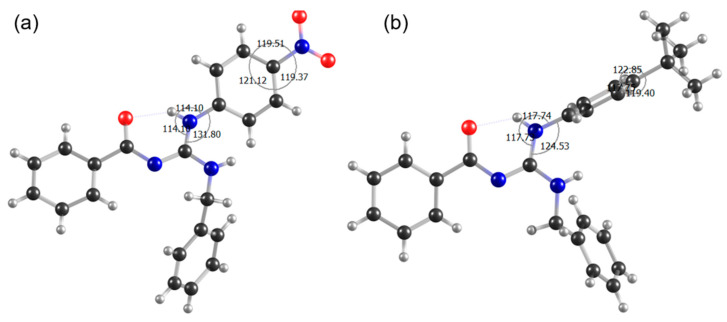
Representation of (**a**) **LQOF-G1** and (**b**) **LQOF-G6** through the crystal structure obtained with XRD with its distortion angle. The second independent molecule in the **LQOF-G6** was omitted for clarity. One intramolecular hydrogen bond of moderate character stabilizes the structure (also valid for the second independent moiety) [24]. Black = carbon, gray = hydrogen, blue = nitrogen, red = oxygen.

**Figure 4 biomolecules-12-01903-f004:**
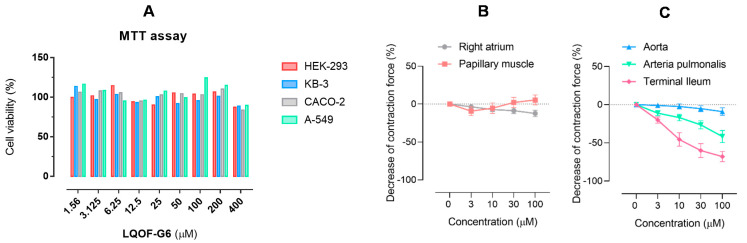
Toxicity results for **LQOF-G6**. (**A**) Bar graph showing the average percentage value of cell viability vs. concentration of **LQOF-G6**. Each measurement was performed in triplicate and the average value is reported. (**B**) Effect of **LQOF-G6** on the spontaneous rate of activity (f) of the guinea-pig right atrium (grey) and the contraction force (f_c_) of the papillary muscle (orange). (**C**) Contraction force (f_c_) of the aorta (blue), the arteria pulmonalis (green), and the terminal ileum (red).

**Figure 5 biomolecules-12-01903-f005:**
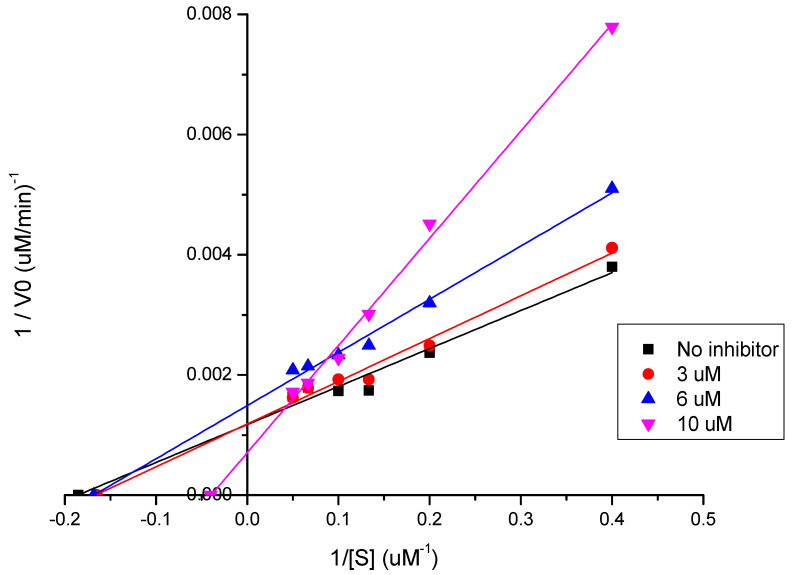
Double reciprocal Lineweaver-Burk plot showing the relationship between the reciprocal of the enzyme activity (1/V_0_) and the reciprocal of the substrate concentration (1/[S]). The intercept of the plot with the abscissa gives the reciprocal of the Michaelis-Menten constant (l/K_m_) in the absence (*w*/*o*) and in the presence of 3, 6, and 10 µM of **LQOF-G6**.

**Figure 6 biomolecules-12-01903-f006:**
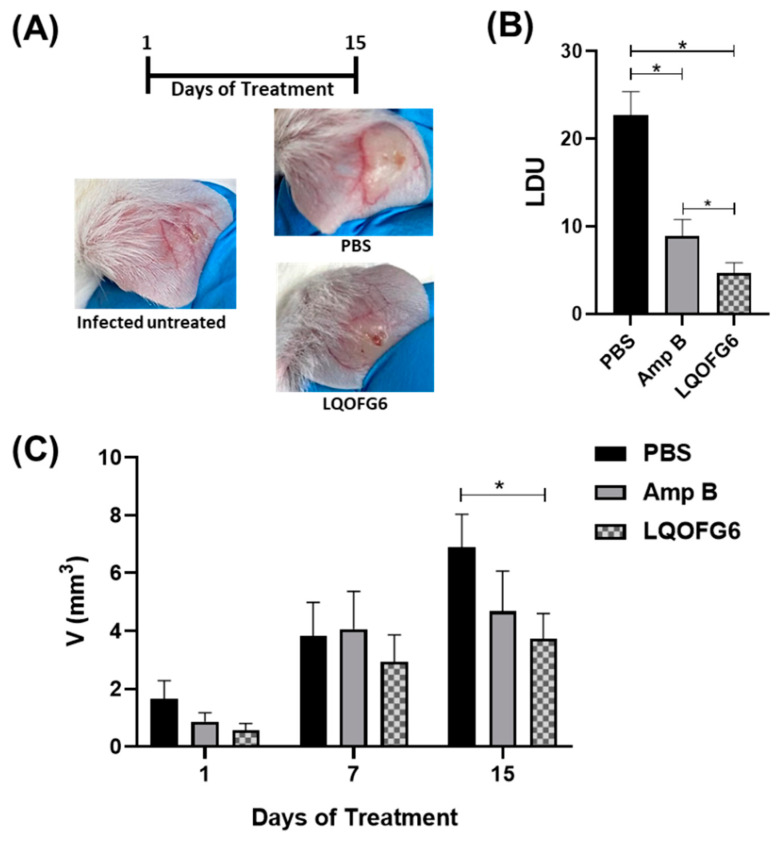
In vivo efficacy of **LQOF-G6** and Amp B treatment in BALB/c mice infected with *L. amazonensis*. (**A**) Representative photographs showing the lesions in the ear of *L. amazonensis*-infected BALB/c mice nontreated or treated with **LQOF-G6** or vehicle (PBS). The infected animals were treated with 2.0 mg/kg/day of **LQOF-G6** or Amp B via IP. (**B**) Parasite load determination: at the end of treatment, the tissue parasite load was quantified using the LDU index (i.e., the number of *Leishmania* amastigotes in 1000 nucleated cells per organ weight). (**C**) Lesion progression during infection: The development of ear lesions was monitored for 15 days and the mean volume (volume (V) = *D* × *d* × *e*, where *D* is the larger diameter, *d* is the minor diameter, and *e* is the thickness) of lesions in each group was calculated. The control used was 1X PBS. Data are expressed as averages plus the SD. * Statistically significant difference between groups (*p* < 0.01).

**Figure 7 biomolecules-12-01903-f007:**
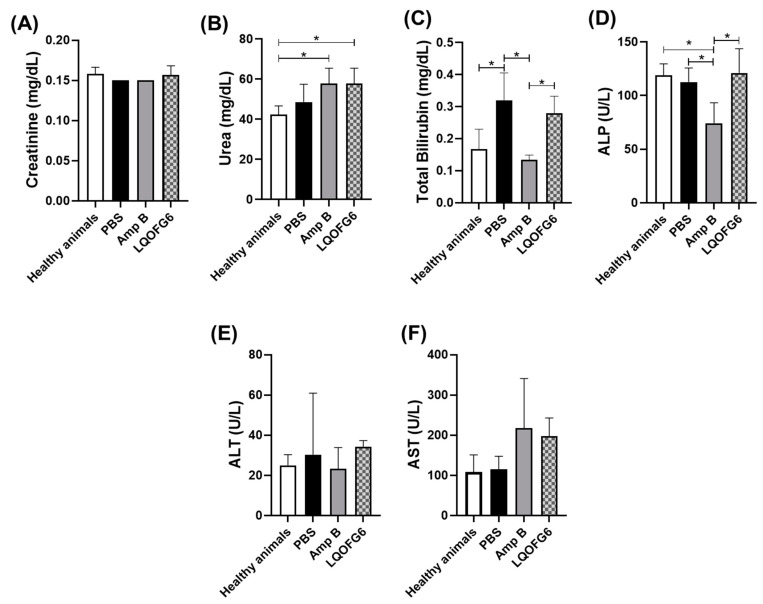
Plasma levels of biomarkers related to hepatic and renal functions in healthy (noninfected) BALB/c mice and animals infected with *L. amazonensis* and treated with either 1X PBS (vehicle control), 2.0 mg/kg/day **LQOF-G6** or 2.0 mg/kg/day Amp B. (**A**) Creatinine levels; (**B**) Urea levels; (**C**) Total bilirubin levels; (**D**) ALP levels; (**E**) ALT levels: (**F**) AST levels. The data are expressed as averages plus SD. * Statistically significant difference between groups (*p* < 0.05).

**Figure 8 biomolecules-12-01903-f008:**
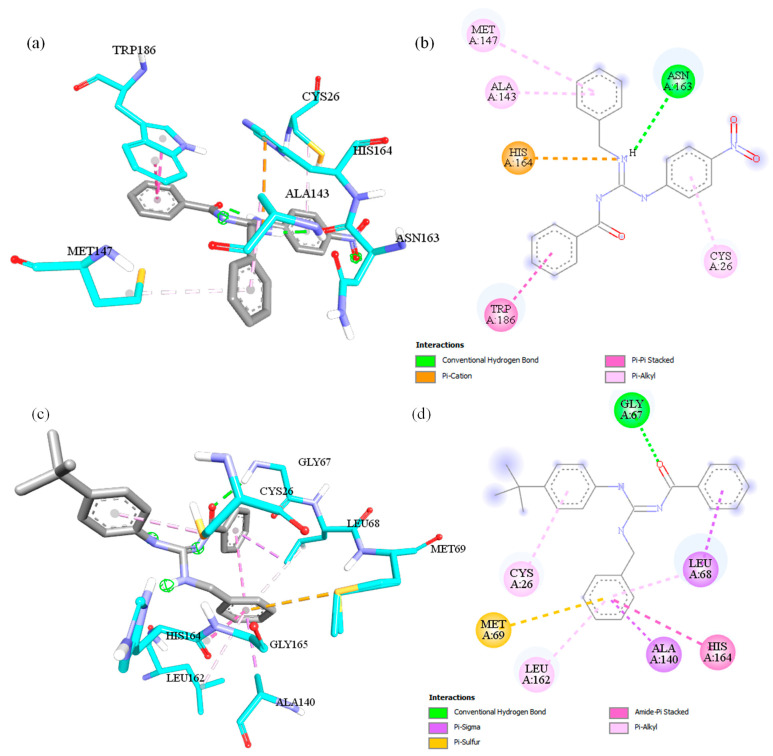
Results of the molecular docking investigation of **LQOF-G1** and interactions representations in (**a**) 3D perspective, (**b**) 2D perspective, and of **LQOF-G6** (**c**) 3D representation of the interactions and (**d**) 2D representation. Cyan = carbon, red = oxygen, purple = nitrogen, yellow = sulfur, and white = hydrogen.

**Table 1 biomolecules-12-01903-t001:** Corresponding groups and treatments for the in vivo leishmanicidal activity test.

Groups	Treatment
1	Infected and treated with PBS + 1% DMSO
2	Infected and treated with Amp B 2 mg/kg/day
3	Infected and treated with **LQOF-G6** 2 mg/kg/day
4	Uninfected and untreated

**Table 2 biomolecules-12-01903-t002:** Experimental parameters and CCDC-Code for compounds **LQOF-G6** and **LQOF-G1**.

Sample	Instrument	Source	Temp.	Detector Distance	Time/Frame	#Frames	Frame Width	CCDC
			[K]	[mm]	[s]		[°]	
**LQOF-G6**	Bruker D8 Venture	Mo	120	40	60	3506	0.3	2,144,089
**LQOF-G1**	Bruker D8 Venture	Mo	100	34	15	4215	0.5	1,992,228

**Table 3 biomolecules-12-01903-t003:** Torsion Angles and bond spatial lengths for **LQOF-G6** and **LQOF-G1** {° (Å)}.

Molecule	Entry	Atoms	Angle/° (A)	Atoms	Length (A)
**LQOF-G6**	1	C2A-N1A-C1A-N3A	0.9 (3)	H(C3A)-H(N3A)	2.90
	2	C1A-N1A-C2A-C3A	81.2(3)	H(C3A)-H(N1A)	3.07
	3	C1A-N1A-C2A-C7A	−98.6(2)	H(C7A)-H(N3A)	3.43
	4	N2A-C12A-C13A-C14A	8.3(3)	H(C7A)-H(N1A)	2.93
**LQOF-G1**	1	C2A-N1A-C1A-N3A	2.9 (3)	H(C3A)-H(N3A)	4.46
	2	C1A-N1A-C2A-C3A	−36.4 (3)	H(C3A)-H(N1A)	2.30
	3	C1A-N1A-C2A-C7A	14.0 (2)	H(C7A)-H(N3A)	1.88
	4	N2A-C12A-C13A-C14A	3.7 (3)	H(C7A)-H(N1A)	3.43

**Table 4 biomolecules-12-01903-t004:** Binding affinity of the “best” mode of the ligand with the active site of 6P4E enzyme and experimental inhibition index.

Ligand	Affinity/kcal.mol^−1^	Inhibition Index/%
**LQOF-G1**	−7.9	-
**LQOF-G2**	−8.3	-
**LQOF-G6**	−8.8	73
**LQOF-G32**	−8.2	53

## Data Availability

Not applicable.

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
