# Peer review of "Novel Selective and Low-Toxic Inhibitor of LmCPB2.8ΔCTE (CPB) One Important Cysteine Protease for Leishmania Virulence"

_biomolecules, 2022, doi:10.3390/biom12121903_

Round 1
Reviewer 1 Report
Most of the comments are attached to the PDF file. some important comments are:
1. Author have to perform the histology to confirm the toxicity effect?
2. Give the table or figure related to animal treatment in antileishmanial activity?

Reviewer 2 Report
1-In the section of synthesis of guanidines authors need to describe the preparation procedure in brief.
2- Did authors see any difference in organ toxicity between Guinea pigs as they used both sexes?
3- How did authors decide to use 2mg/kg of LQOF-G6 for treatment of BALB/C mice?
4- Figure 7 needs legend and graphs are not shown well.
5- In line 267, 107 (Inoculated promastigotes), 7 need to be edited to superscript.
